# Sonographic measurement of normal common bile duct diameter and associated factors at the University of Gondar comprehensive specialized hospital and selected private imaging center in Gondar town, North West Ethiopia

**Misganaw Gebrie Worku**[1]*, **Engdaw Fentahun Enyew**[1], **Zerubabel Tegegne Desita**[2], **Abebe Muche Moges**[1]

1 Department of Human Anatomy, University of Gondar, College of Medicine and Health Science, School of Medicine, Gondar, Ethiopia, 2 Department of Radiology, University of Gondar, college of Medicine and Health Science, School of Medicine, Gondar, Ethiopia

* misgeb2008@gmail.com

## Abstract

### Background

The biliary tree is a three-dimensional system of channels that bile flows from the hepatocytes to the gallbladder and to the intestine. Size of the common bile duct (CBD) is a predictor of biliary obstruction and, therefore, its measurement is an important component of biliary system evaluation. Factors like age, height, weight, BMI, previous cholecystectomy, drugs, and type of imaging modality affect CBD diameter, but the duct significantly dilated due to obstructive biliary pathology.

### Objective

To measure the normal CBD diameter and its association with age, sex, and anthropometric measurement at the University of Gondar Comprehensive Specialized Hospital and selected private imaging center, Gondar town, Northwest Ethiopia, 2019.

### Methods and materials

Institutional based cross-sectional study was conducted on 206 subjects without any history of hepatobiliary abnormality. The CBD measured at the proximal part just caudal to the porta hepatis. Descriptive analysis, student t-test, one way ANOVA, correlation and both bivariable and multivariable linear regression analysis were implemented. In bivariable linear regression variables with p-value, less than 0.2 were selected for multivariable analysis and in multivariable linear regression analysis variables with P-Value less than 0.05 were considered as statistically significant.

**Data Availability Statement:** All relevant data are within the manuscript and its Supporting Information files.

**Funding:** To conduct this study the author receive funding from University of Gondar, but the university had no role in study design, data collection and analysis, decision to publish or preparation of the manuscript.

**Competing interests:** The authors have declared that no competing interests exist.

**Abbreviations:** CBD, common bile duct; BMI, body mass index; SD, standard deviation; UOG, University of Gondar.

## Results

The mean age of the study participants was 39.4 (range 18–87). The mean diameter of the CBD was 3.64mm 95%CI (3.52, 3.77), which ranges from 1.8 to 5.9 mm, with 65% of the participant having CBD diameter less than 4mm. The diameter of CBD significantly associated with age with a linear trend. The mean diameter in a rural area was greater than subjects living in an urban area. Independent t-test showed no statistically significant difference in CBD diameter between male and female subjects.

## Conclusion

The lower limit of the CBD diameter for this study was similar to most of the studies, but the upper limit was found to be slightly lower. The diameter was significantly associated with age along the linear trend and it was progressively increased from the lower age group onwards. The diameter of CBD did not show statistically significant association with any of the anthropometric measurement.

## Introduction

The Common bile duct (CBD) is approximately 7 cm long and can vary from 5 to 15 cm. This variation depending on the site where the cystic duct joins the common hepatic duct and its diameter is usually around 6 mm in adults[1, 2]. It leaves the lesser omentum and descends posterior to the first part of the duodenum and then penetrates the posterior part of the head of the pancreas [3]. Anatomically the CBD lies anteriorly and to the right of the portal vein and hepatic artery in the porta hepatis and gastrohepatic ligament. This constant relationship helps to demonstrate CBD and differentiate it from the portal vein and hepatic artery by using ultrasonography [1, 4]

The upper limit of the normal common bile duct (CBD) diameter is controversial because in addition to biliary obstruction factors like age, height, weight, BMI, previous cholecystectomy, drugs, and the imaging modality itself can affect its diameter measurement [3, 4, 5, 6].

Diseases and biliary disorders associated with biliary system obstruction affect a significant portion of the world's population[7]. The size of the CBD is a predictor of this obstruction and an important component of biliary system evaluation. Prior knowledge on the internal measurements of the CBD diameter distinguished obstructive cause of jaundice from non-obstructive causes [5, 7] and the finding of abnormally dilated CBD is the most common indicators of choledochostomy. To evaluate the importance of CBD, it is necessary to know about the normal variations of CBD diameter[8, 9]. The assessment of common bile duct also helps for the evaluation of strictures or filling defects of hepatobiliary system [10].

The common bile duct diameter increased due to age, but its diameter significantly dilated as a result of obstructive biliary pathology. Studies had report the normal upper limit of CBD diameter measured with ultrasonography was 6mm or less [3, 5, 11], but other studies had documented the upper limit was greater than 7mm[7, 12, 13, 14]. So in order to diagnosis and manage pathology of the hepatobiliary system, a standard reference value on the normal measurement of the internal diameter of the common duct is very important [15, 16].

Even if the knowledge of the normal standard reference of the CBD diameter is crucial in the diagnosis and management of the biliary system pathology, there is not enough study for its standard measurement done in our population. There are many studies which try to assess

the normal range of diameter of CBD, but many of the studies did not asses the normal value at different age and BMI groups [16]. Also most of studies done in this thematic area didn't consider the methodological issue including sample selection and sample size calculation, so this study take under consideration of this issue.

There is no enough study for the standard measurement of CBD diameter among the Ethiopian population. So this study aims to assess the range of normal measurements of CBD diameter and its association with age, sex, and anthropometric measurement.

## Methods and materials

### Study area and period

Institutional based cross-sectional study design was conducted at the University of Gondar comprehensive specialized hospital and selected private imaging center in Gondar town from January to February 2019.

### Sample size determination and sampling technique

The sample size was calculated using the formula designed for continues data by taking the standard deviation and margin of error of similar study (SD = 1.02 and margin of error = 0.141 with 95% CI) [7]; Therefore the sample size $n = \frac{Za/2^2 \delta^2}{d^2}$, with δ = 1.02 and d = 0.141 [7]

n = (1.96)$^2$(1.02)$^2$ / (0.141)$^2$

n = 200.9 = 201 subject, 2% non-response rate from other study [13], so 2/100× 201 = 4.02

Hence, in the present study a total of 206 subject were participated.

In the present study, a systematic random sampling method was implemented. The total population was estimated by considering the patient who came for an ultrasound investigation per day and a total of on average around 70 patients underwent an ultrasound investigation as an outpatient department. The interval was calculated as (70×26)/206 = 8.8. Among the first eight participants, one was selected randomly and sampling includes every 8 patients until the targeted sample size was achieved. A total of 206 subjects comprising normal healthy individuals or subjects presenting to the hospital with medical conditions other than those of hepato-biliary systems and non-pregnant women visiting the hospital for regular checkups were involved.

### Study variables

Dependent variable; diameter of the common bile duct, independent variables; include Socio-demographic data age, sex, place of residence and anthropometric characteristics height, weight, and body mass index.

### Data collection tools and procedures

Ethical clearance were obtained from the ethical review committee of the School of Medicine, University of Gondar and informed verbal consent obtained from each individual at the time of data collection. Socio-demographic characteristics related to age, sex, and place of residence were properly recorded for each subject. Ultrasonographic findings with regard to common bile duct diameter were obtained. In order to reduce observer bias, single expert radiologist was involved in conducting ultrasonography for all subjects. The diameter was measured at the proximal part of the duct after assessing the entire duct for the presence of any abnormality that affects CBD diameter. The measurement was done either in the longitudinal or transverse view and most measurements in this study were done with longitudinal view and diameter

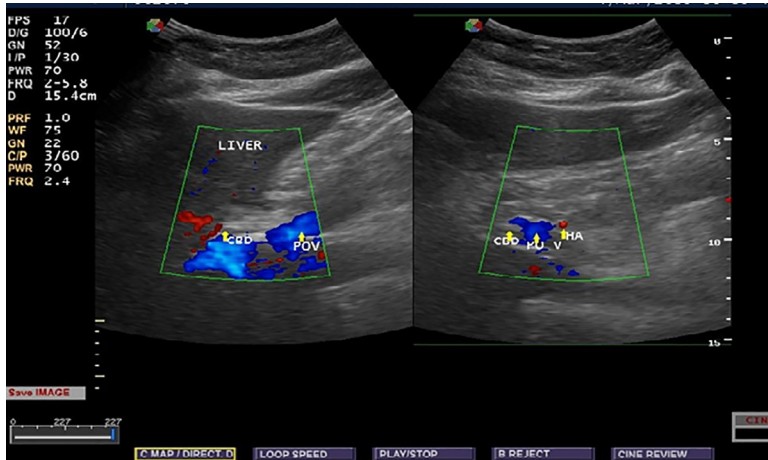

**Fig 1. Longitudinal and transverse view showing the relation of the common bile duct to the portal vein and hepatic artery (POV -portal vein, CBD common bile duct, HA hepatic artery).**

measured perpendicular to its long axis using the standard sonographic measurement techniques (Figs 1 and 2).

The study subjects were fasted for around 4 to 6 hours and the measurement was taken at deep inspiration. History and physical examination were undertaken to exclude patients with disease condition that affects CBD and on ultrasonography patients with cholecystectomy, cholelithiasis, choledocholithiasis, cholecystitis, cholangitis and any hepatobiliary and pancreatic abnormality or mass that affect the diameter were excluded from the study.

## Data processing and analysis

The collected data were checked for completeness, accuracy, and clarity before analysis. The data were entered into Epi- info version 7.2.0.1 and transferred to SPSS version 20 for analysis. The results were presented in the form of tables, figures, graph and text using frequencies and

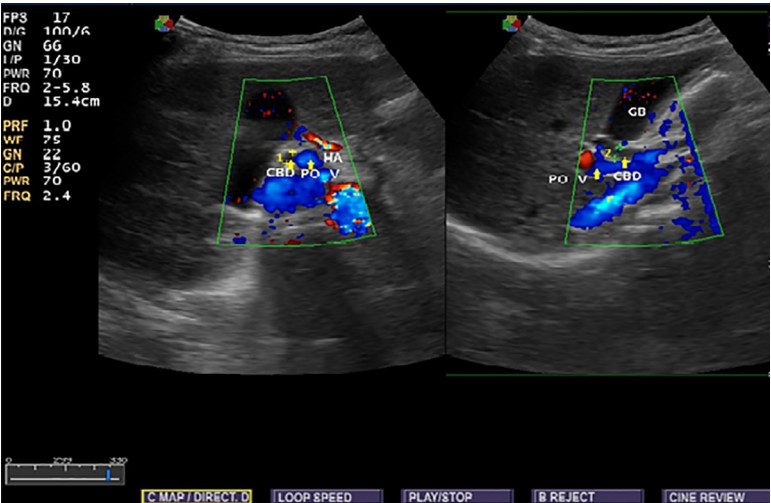

**Fig 2. Transverse and longitudinal view showing the measurement of the diameter of the common bile duct at the portal hepatics (POV-portal vein, CBD common bile duct, HA hepatic artery).**

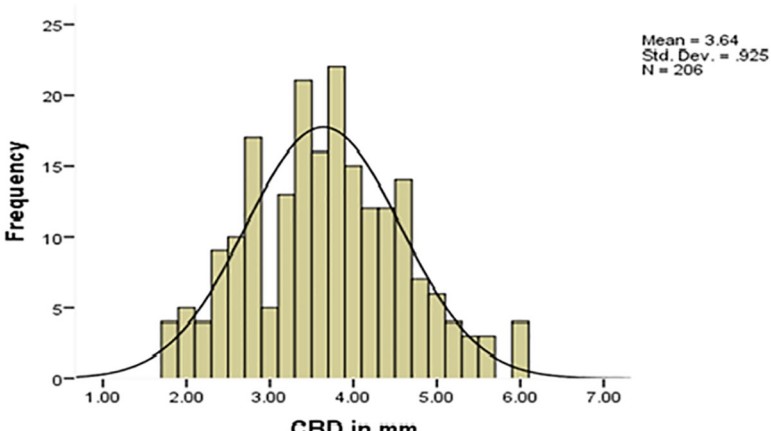

**Fig 3. Histogram showing the frequency distribution of common bile duct diameter.**

summary statistics. The means (± standard deviation), ranges, minimum, maximum value and the 95% confidence intervals for the mean were calculated. The Levene's Test of homogeneity of Variances, Kolmogorov-Smirnov and Shapiro-Wilk test of normality, multicollinearity diagnosis, and other assumptions were checked before doing any statistical analysis and were fulfilled and the distribution of the data were also assessed using histogram (Figs 3 and 4). In order to assess the continuous dependent variable CBD diameter at different age groups and body mass index, ANOVA test was done.

The relationship between the diameter of the CBD and each of the variables was assessed with Pearson's correlation coefficient and both bivariable and multivariable linear regression analysis were done. At bivariable linear regression analysis variables with p-value, less than 0.2 were selected for multivariable analysis and at multivariable linear regression analysis, the variables with P-Value less than 0.05 were considered as statistically significant. Differences of continuous variables between two independent groups were assessed with the 2-tailed independent sample t-test.

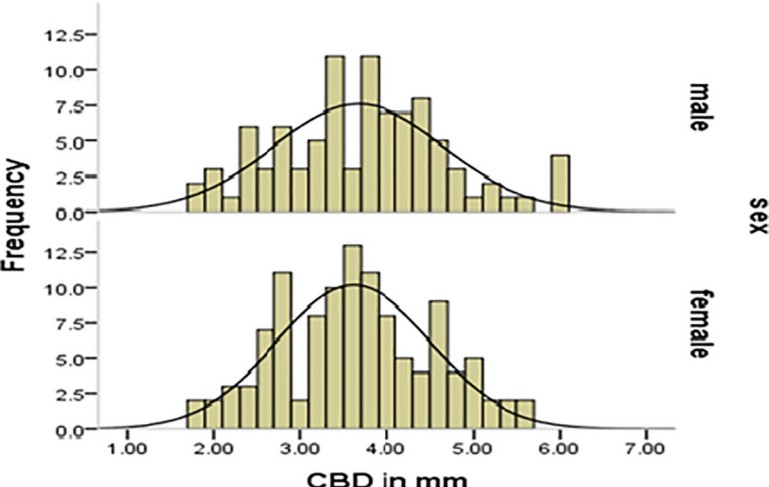

**Fig 4. Histogram showing the frequency distribution of common bile duct diameter by gender.**

## Ethical issue

Ethical clearance was obtained from the ethical review committee of the School of Medicine, College of Medicine and Health Sciences, University of Gondar. An official letter was submitted to the University of Gondar Comprehensive Specialized Hospital Radiology Department and to the selected private imaging center and respective official permission was obtained. The purpose and importance of the study were explained to the study subjects. Confidentiality was maintained at all levels of the study and informed verbal consent obtained from each individual at the time of data collection.

## Results

A total of 206 subjects were studied with unequal proportion with respect to sex. The study subjects were in the age group of 18–87 years and the mean age was 39.4 years (SD 16.4 years). A majority of the participant belongs to the age group of 18–27 years. The mean age for males was 41.4 years and females 37.8 years. The mean weight, height, and BMI of the participants were 57.8 kg (SD 11.5 kg), 1.63m (SD 0.086m) and 21.6 KG/M$^2$ respectively.

The mean diameter measured at the proximal part of the CBD was 3.64(SD 0.92) mm. The normal range of CBD diameter was 1.8mm to 5.9 mm. However, 65% of the study participants had a common bile duct diameter of < 4 mm.

The majority of the patients irrespective of the age group had the diameter which ranges from 2–3.9 mm. But, most subjects in the old age group had a CBD diameter ranges from 4–5.9mm.

Most of the participants were in the BMI category of 18–24.9 Kg/m$^2$ and the majority of the subjects in this category had a diameter which ranges from 2–3.9 mm. On the other hand, most study subjects with BMI of 30–39.9 Kg/m$^2$ had CBD diameter range of 4–5.9 mm (Tables 1–3).

But, the mean diameter of CBD didn't show a statistically significant difference among various categories of BMI (p>0.05) (Fig 6). Post hoc (Bonferroni) test was done to identify in which age groups had the mean CBD diameter difference found.

Bonferroni test showed that there was a significant mean difference in CBD diameter of the age group of 18–27 years as compared to the age group of 48–57 years and ≥ 58 years. Similarly, there was a mean diameter difference in the age group of 28–37 years with the age group of 48–57 years and age ≥ 58 years. The mean CBD diameter in age group of 38–47 years had also a significant difference with the age group of 48–57 years and age ≥ 58 years old. However, no mean difference in the diameter of the CBD between the age group of 48–57 years and age group of ≥ 58 years (Table 4).

The mean diameter of the common bile duct was observed to be 3.95 (SD 1.001 mm) for rural and 3.4 mm (SD 0.79 mm) for urban. This difference was tested by independent samples t-test, which was statistically significant (p ≤0.001). On the contrary, there was no significant difference in mean CBD diameter between the sex of study subjects (Fig 5) (p>0.05).

**Table 1. Diameter of CBD stratified by age group, Northwest Ethiopia, 2019.**

| CBD in mm | Age in (years) n (%) | | | | | Total |
|---|---|---|---|---|---|---|
| | 18–27 | 28–37 | 38–47 | 48–57 | ≥58 | |
| <2 | 1(1.78) | 2(3.6) | 1(3.2) | 0(0) | 0(0) | 4 |
| 2–3.9 | 44(78.5) | 39(70.9) | 23(74.2) | 11(35.5) | 13(39.4) | 130 |
| 4–5.9 | 11(19.6) | 14(25.4) | 7(22.5) | 20(64.5) | 20(60.6) | 72 |
| Total | 56(100) | 55(100) | 31(100) | 31(100) | 33(100) | 206 |

**Table 2. Diameter of CBD stratified by sex, Northwest Ethiopia, 2019.**

| CBD (mm) | Sex of respondent | | Total |
|---|---|---|---|
| | Male | Female | |
| <2 | 2(2.2) | 2(1.8) | 4 |
| 2–3.9 | 55(59.1) | 75(64.4) | 130 |
| 4–5.9 | 36(38.7) | 36(31.9) | 72 |
| Total | 93(100) | 113(100) | 206 |

In both sexes, there was a significant positive correlation between the diameter of the CBD and age (p<0.001, r = 0.41). However, no statistically significant correlation between the diameter of the CBD and weight, height and BMI (Table 5).

In linear regression analysis variables with p-value, less than 0.2 in the bivariable analysis were selected for multivariable linear regression analysis. Bivariable linear regression analysis of age, sex, weight, height, and residency with CBD diameter were done separately and only the age and residency had a p-value less than 0.2 (Table 6). In the multivariable linear regression analysis, age and residency were also significantly associated with common bile duct diameter (p<0.001). Keeping other variables constant, one year increase in the age of respondents increases the diameter by the factor of 0.021. The diameter of CBD decreased by the factor of 0.42 in urban as compared to rural population (Table 7).

## Discussion

The sonographic measurement of common bile duct via ultrasonography is a cheap, non-invasive and easily available means of evaluating the hepatobiliary system and other abdominopelvic organs. In many parts of Ethiopia and other resource-limited countries, ultrasonography may be the only available method.

In the present study, the mean diameter of CBD was 3.64mm with 95% CI (3.52, 3.77) which was consistent with the study conducted in Iran (mean = 3.64) [8]. This finding was smaller than many findings[1, 5, 6, 7, 16]. However higher than studies conducted in British and Israel reported 3.2mm and 3.39mm, respectively, [14]. This difference may be due to the nutritional variation and/or maybe related to the respiratory phase of the respondent at the time of measurement as the diameter of the CBD slightly increased during deep inspiration [17]. The lower and the upper limit of normal CBD diameter in this study were 1.8mm and 5.9mm, respectively. However, the majority of the study subjects (65%) had a common bile duct diameter of < 4mm. The upper limit was similar to a report in Khartoum(6.1mm) and Addis Ababa(6mm) [3, 5]. Study in British found the upper limit 4 mm which was smaller

**Table 3. Diameter of CBD stratified by body mass index, Northwest Ethiopia, 2019.**

| CBD(mm) | Body mass index | | | | Total |
|---|---|---|---|---|---|
| | <18KG/M$^2$ | 18–24.9 | 25–29.9 | 30–39.9 | |
| <2 | 0(0) | 3(2.3) | 1(2.7) | 0(0) | 4 |
| 2–3.9 | 23(62.2) | 81(63.3) | 25(67.6) | 1(25) | 130 |
| 4–5.9 | 14(37.8) | 44(34.4) | 11(29.7) | 3(75) | 72 |
| Total | 37(100) | 128(100) | 37(100) | 4(100) | 206 |

The mean diameter increase from 3.2 mm among those aged 18–27 years to 4.2 mm in the age group of more than 57 years. The average diameter of the common bile duct was compared by using the one way ANOVA test in the different age groups and a significant difference was found(p<0.05) (Fig 5).

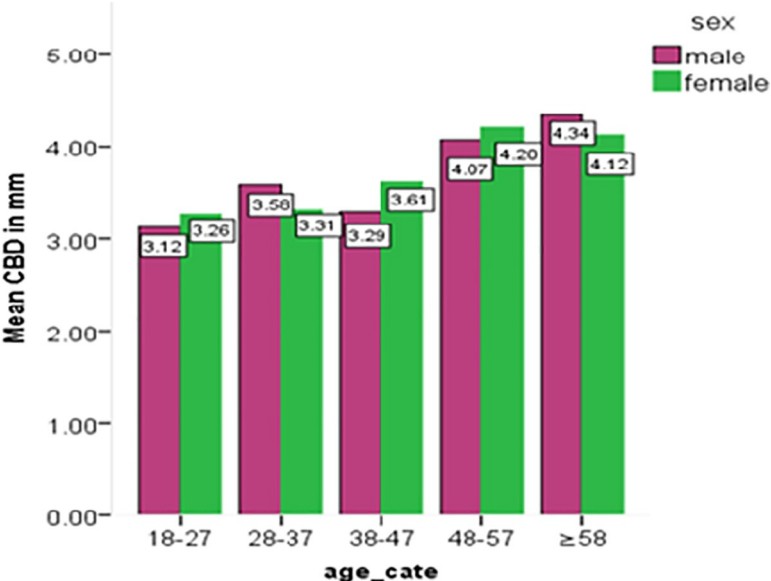

**Fig 5. Bar chart showing the mean common bile duct diameter at different age group by sex.**

than our finding [18]. However, many studies reported that the upper limit of the CBD diameter was 8.6mm, 9mm, and 10mm, which was higher than our finding. [8, 11, 12].

A significant positive association was observed between the diameter of the CBD and age. As age increases the diameter of CBD increases, which was supported by many studies [4, 5, 6, 7, 11]. However, a report in Philadelphia and Nigeria showed that no statistically significant association between the diameter of CBD and age[12, 13]. This may be due to a variety of ultrasound machines used for the measurement. In this study, we didn't find any statistical association of common bile duct diameter with sex. This finding was similar to other studies by

**Table 4. Post hoc (Bonferroni) test done for mean CBD diameter difference at different age groups, Northwest Ethiopia, 2019.**

| Age_cate(I) | age_cate(J) | Mean Difference (I-J) | Sig. | 95% Confidence Interval | |
| --- | --- | --- | --- | --- | --- |
| | | | | Lower Bound | Upper Bound |
| 18–27 | 28–37 | -0.26 | 1.000 | -0.71 | 0.18 |
| | 38–47 | -0.30 | 1.000 | -.83 | 0.23 |
| | 48–57 | -0.94 | 0.000 | -1.47 | -0.40 |
| | ≥58 | -1.03 | 0.000 | -1.56 | -0.51 |
| 28–37 | 18–27 | 0.26 | 1.000 | -0.18 | 0.71 |
| | 38–47 | -0.03 | 1.000 | -0.57 | 0.49 |
| | 48–57 | -0.67 | 0.004 | -1.21 | -0.143 |
| | ≥58 | -0.77 | 0.000 | -1.29 | -0.24 |
| 38–47 | 18–27 | 0.30 | 1.000 | -0.23 | 0.83 |
| | 28–37 | 0.038 | 1.000 | -0.49 | 0.57 |
| | 48–57 | -0.64 | 0.030 | -1.24 | -0.03 |
| | ≥58 | -0.74 | 0.006 | -1.33 | -0.13 |
| 48–57 | 18–27 | 0.94 | 0.000 | 0.41 | 1.47 |
| | 28–37 | 0.67 | 0.004 | 0.14 | 1.21 |
| | 38–47 | 0.64 | 0.030 | 0.03 | 1.24 |
| | ≥58 | -0.09 | 1.000 | -0.69 | 0.50 |

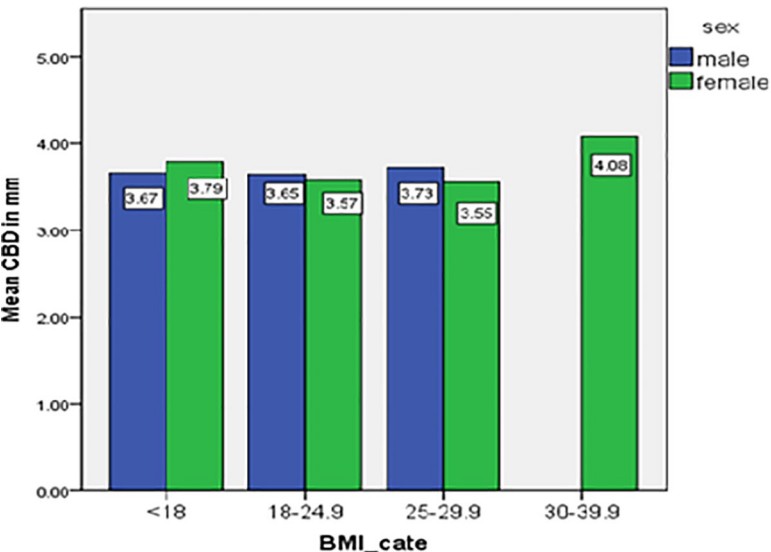

**Fig 6. Bar graph showing the mean common bile duct diameter at different BMI categories.**

**Table 5. Mean and standard deviation of common bile duct diameter by age group.**

| Age group | Frequency | Mean | Std. deviation |
|---|---|---|---|
| 18–27 | 56 | 3.2 | 0.76 |
| 28–37 | 55 | 3.4 | 0.75 |
| 38–47 | 31 | 3.5 | 0.74 |
| 48–57 | 31 | 4.1 | 1.05 |
| $\geq$ 58 | 33 | 4.2 | 0.95 |
| Total | 206 | 3.6 | 0.92 |

**Table 6. Summary of correlation between common bile duct diameter, age and anthropometric measurements by sex.**

| Variable | Male | Female |
|---|---|---|
| | Pearson's correlation(r) Sig (2-tailed) | Pearson's Sig (2-tailed) correlation(r) |
| Age | 0.409 0.00 | 0.42 0.00 |
| Weight | -0.03 0.97 | 0.02 0.81 |
| Height | -0.09 0.39 | 0.4 0.67 |
| BMI | 0.052 0.61 | 0.007 0.93 |

**Table 7. Multiple linear regression of CBD diameter with age and residency, Northwest Ethiopia, 2019.**

| Variable | Coefficient | Std.error | T | Sig. | 95.0% Confidence Interval | |
|---|---|---|---|---|---|---|
| | | | | | Lower Bound | Upper Bound |
| Constant | 3.06 | .17 | 17.34 | 0.000 | 2.711 | 3.406 |
| Age | .021 | .004 | 5.9 | 0.000 | 0.014 | 0.028 |
| Urban residency | -.420 | .11 | -3.55 | 0.000 | -0.653 | -0.187 |

Admassie [5], Adibi and Givechian [8]. The current study showed that common bile duct diameter had no statistically significant association with any of the anthropometric measurements. Admassie[5] and Mohammad SH [13] found a significant positive correlation with weight, but no with height and body mass index. Mohammad [3]also found a significant correlation of common bile duct diameter with height, weight, and body mass index.

## Limitation of the study

In this study chemical analysis were not done primarily rather reviewed from the chart and patient having positive result excluded and in this study patients having hepatobiliary abnormality that had an effect on our study were excluded by detail history and physical examination, reviewing the medical history including any chemical analysis done from the chart and finally through ultrasonographic investigation by a single senior radiologist. Also most of the study participant in this study are old age which may affect our result, as common bile duct diameter increased by around 1mm each decade in the old age groups.

## Conclusion

The lower limit of CBD diameter for this study was similar to most of the studies conducted, but the upper limit was found slightly lower. The diameter of the CBD was significantly associated with age along the linear trend. There was a mean common bile duct diameter difference among various age groups, but not in different categories of body mass index. Its mean value was higher among rural as compared to urban population, however no mean CBD diameter difference between sex. The diameter of the CBD did not show a statistically significant association with the anthropometric measurement.

## Supporting information

**S1 File. Confidentiality and informed consent statement.**
(DOCX)

**S2 File. English version of data collection check list.**
(DOCX)

**S3 File. Ethical clearance form.**
(DOCX)

## Acknowledgments

We are grateful to thank the study participant for their valuable contribution and provide appropriate information. The authors like to express their gratitude to all the members of the Department of Human Anatomy as well as the Radiology department of the University of Gondar comprehensive specialized hospital and selected private imaging center in Gondar town as their contributions were vital in the completion of this research work.

## Author Contributions

**Conceptualization:** Misganaw Gebrie Worku, Engdaw Fentahun Enyew, Zerubabel Tegegne Desita, Abebe Muche Moges.

**Data curation:** Misganaw Gebrie Worku.

**Formal analysis:** Misganaw Gebrie Worku.

**Funding acquisition:** Misganaw Gebrie Worku.

**Investigation:** Misganaw Gebrie Worku.

**Methodology:** Misganaw Gebrie Worku, Zerubabel Tegegne Desita.

**Project administration:** Misganaw Gebrie Worku.

**Resources:** Misganaw Gebrie Worku, Zerubabel Tegegne Desita.

**Software:** Misganaw Gebrie Worku.

**Supervision:** Misganaw Gebrie Worku, Engdaw Fentahun Enyew, Zerubabel Tegegne Desita, Abebe Muche Moges.

**Validation:** Misganaw Gebrie Worku.

**Visualization:** Misganaw Gebrie Worku, Engdaw Fentahun Enyew, Zerubabel Tegegne Desita, Abebe Muche Moges.

**Writing – original draft:** Misganaw Gebrie Worku, Engdaw Fentahun Enyew, Zerubabel Tegegne Desita, Abebe Muche Moges.

**Writing – review & editing:** Misganaw Gebrie Worku, Engdaw Fentahun Enyew, Zerubabel Tegegne Desita, Abebe Muche Moges.

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
