## [Editor Report · Decision Letter 0]

1 Oct 2019

PONE-D-19-22390

Sonographic Measurement of normal Common Bile Duct Diameter and its correlation with Age, Sex and Anthropometric measurement at University of Gondar comprehensive hospital and selected private imaging center in Gondar town, North West Ethiopia.

PLOS ONE

Dear Mr Worku,

Thank you for submitting your manuscript to PLOS ONE. After careful consideration, we feel that it has merit but does not fully meet PLOS ONE’s publication criteria as it currently stands. Therefore, we invite you to submit a revised version of the manuscript that addresses the points raised during the review process.

We would appreciate receiving your revised manuscript by Nov 15 2019 11:59PM. To enhance the reproducibility of your results, we recommend that if applicable you deposit your laboratory protocols in protocols.io, where a protocol can be assigned its own identifier (DOI) such that it can be cited independently in the future. For instructions see: http://journals.plos.org/plosone/s/submission-guidelines#loc-laboratory-protocols

We look forward to receiving your revised manuscript.

Kind regards,

Roberto Coppola, MD, FACS

Academic Editor

PLOS ONE

Journal Requirements:

1. Please amend either the title on the online submission form (via Edit Submission) or the title in the manuscript so that they are identical.

2. We note you have included a table to which you do not refer in the text of your manuscript. Please ensure that you refer to Tables 6 & 7 in your text; if accepted, production will need this reference to link the reader to the Table.

3. Please ensure that you refer to Figures 1-6 in your text as, if accepted, production will need this reference to link the reader to the figure.

Additional Editor Comments (if provided):

This is a very interesting paper. The knowledge of the variation of the CBD diameter according to the age of the subject is not new. But the Authors have investigated this simple but crucial data in a very thorough modality. Thy have measured by US the common bile duct of 206 subjects enrolled in a precise statistical modality. The subjects were normal and investigated at the Hospital for other reasons than biliary diseases. The majority of the subjects were young, maybe according to the age population of this district. The sample size estimation was correct. They have also performed a correct US examination with precise landmarks before measure the CBD size. They have tried to minimize any measurement errors.

The analysis of the data is very accurate. The results are reported with clear explanation in multiple tables and graphs. In the "discussion" paragraph many previous studies on this issue are reported and the data are compared. Interesting the look at the demographic and anthropologic aspect of the evolution of the CBD in their Region. The conclusions are fully supported by the data.

The paper is very easy to read, interesting and I think useful for our readers.

I suggest to add at the and of the Discussion a comment on the limitations of this study, considering the age of the subjects, the technical aspect of this type of measure and else.

One final question for the Authors: do you have performed chemical analysis (liver functional test) of these subjects to completely rule out any liver alteration?
---

## [Author Response · Author response to Decision Letter 0]

19 Oct 2019

Dear PLOS ONE editors and reviewers,

We would like to thank for sharing their views and novel scholarly experiences. The comments are very imperative which we strongly believe in improving the manuscript. The point-by-point responses for each of the comments, questions, and the revised manuscript is provided in the attached documents.

Editors and reviewers comments, 

1. Please amend either the title on the online submission form (via Edit Submission) or the title in the manuscript so that they are identical.

2. We note you have included a table to which you do not refer in the text of your manuscript. Please ensure that you refer to Tables 6 & 7 in your text; if accepted, production will need this reference to link the reader to the Table.

3. Please ensure that you refer to Figures 1-6 in your text as, if accepted, production will need this reference to link the reader to the figure.

4. Do you have performed chemical analysis (liver functional test) of these subjects to completely rule out any liver alteration?

Author’s response,

 1. The comment accepted and the title amended on the online submission and become identical with the title of the main manuscript. The figure files also uploaded to the Preflight Analysis and Conversion Engine (PACE) digital diagnostic tool as TIFF form. 

2. Dear PLOSE ONE editorial the comment question raised by your editorial accepted the table and figures which were not previously referred now referred and included in the full text. 

3. The file previously included (declaration part) which is out of PLOSE ONE format also excluded in the revised manuscript.

4. Dear reviewers, in our study chemical analysis were not done rather reviewed from the chart and patient having positive results were primarily excluded since it is usually done as a base line investigation. In addition patients having hepatobiliary abnormality that had an effect on our study are excluded by detail history, physical examination and reviewing the medical history including any chemical analysis done and finally through ultrasonographic investigation by a senior radiologist. So Patients having positive results were primarily excluded from our study participants and patients having no chemical analysis like liver function test as a baseline investigation were appropriately examined. During ultrasound investigation patients having positive findings associated with the hepatobiliary system were also excluded. After all this process and exclusion of patients having problem related to the hepatobiliary system, the common bile duct diameter was measured by a single radiologist.

---

## [Decision Letter · Decision Letter 1]

13 Dec 2019

Sonographic Measurement of normal Common Bile Duct Diameter and associated factors at the University of Gondar Comprehensive Specialized Hospital and selected private imaging center in Gondar town, North West Ethiopia.

PONE-D-19-22390R1

Dear Dr. Worku,

We are pleased to inform you that your manuscript has been judged scientifically suitable for publication and will be formally accepted for publication once it complies with all outstanding technical requirements.

With kind regards,

Roberto Coppola, MD, FACS

Academic Editor

PLOS ONE

Additional Editor Comments (optional):

The Authors have added a paragraph concerning the limitations of the study. The content of this paragraph is sufficient according to previous request. But the entire paragraph must be reviewed for the English language.
---

## [Editor Report · Acceptance letter]

9 Jan 2020

PONE-D-19-22390R1 

Sonographic Measurement of normal Common Bile Duct Diameter and associated factors at the University of Gondar Comprehensive Specialized Hospital and selected private imaging center in Gondar town, North West Ethiopia. 

Dear Dr. Worku:

I am pleased to inform you that your manuscript has been deemed suitable for publication in PLOS ONE. Congratulations! Your manuscript is now with our production department. 

With kind regards,

on behalf of

Professor Roberto Coppola 

Academic Editor

PLOS ONE